# TAKEN: A Traffic Knowledge-Based Navigation System for Connected and Autonomous Vehicles

**DOI:** 10.3390/s23020653

**Published:** 2023-01-06

**Authors:** Nikhil Kamath B, Roshan Fernandes, Anisha P. Rodrigues, Mufti Mahmud, P. Vijaya, Thippa Reddy Gadekallu, M. Shamim Kaiser

**Affiliations:** 1Department of Computer Science and Engineering, NMAM Institute of Technology, NITTE (Deemed to be University), Nitte 574110, India; 2Department of Computer Science, Computing and Informatics Research Centre, and Medical Technologies Innovation Facility, Nottingham Trent University, Clifton Lane, Nottingham NG11 8NS, UK; 3Department of Mathematics and CS, Modern College of Business and Science Bowshar, Muscat 133, Oman; 4Department of Information Technology Vellore Institute of Technology, Vellore 632014, India; 5Department of Electrical and Computer Engineering, Lebanese American University, Byblos 13-5053, Lebanon; 6Institute of Information Technology, Jahangirnagar University, Dhaka 1342, Bangladesh

**Keywords:** connected and autonomous vehicles, deep learning, traffic knowledge sharing, navigation

## Abstract

Connected and autonomous vehicles (CAVs) have witnessed significant attention from industries, and academia for research and developments towards the on-road realisation of the technology. State-of-the-art CAVs utilise existing navigation systems for mobility and travel path planning. However, reliable connectivity to navigation systems is not guaranteed, particularly in urban road traffic environments with high-rise buildings, nearby roads and multi-level flyovers. In this connection, this paper presents TAKEN-Traffic Knowledge-based Navigation for enabling CAVs in urban road traffic environments. A traffic analysis model is proposed for mining the sensor-oriented traffic data to generate a precise navigation path for the vehicle. A knowledge-sharing method is developed for collecting and generating new traffic knowledge from on-road vehicles. CAVs navigation is executed using the information enabled by traffic knowledge and analysis. The experimental performance evaluation results attest to the benefits of TAKEN in the precise navigation of CAVs in urban traffic environments.

## 1. Introduction

Connected and autonomous vehicles (CAVs) are emerging as next-generation transportation in terms of growing connectivity and automation devices in vehicles [1,2]. Most modern cars have a dashboard and connectivity to the internet or personal devices enabling a wide range of traffic services for easing driving and travel for drivers [3]. Nowadays, the majority of the drivers use GPS service travel pathfinder, which helps in reaching the correct destination particularly significant in urban dense road networks [4,5]. Internet connectivity enabled traffic services to not only improve safety and travel experience but also resulted in green transport, considering lesser congested and shorter travel path-oriented guidance to drivers [6]. The green transport capability of modern vehicles has attracted government attention around the work towards supporting CAV technology development and implementation in coming years [7,8]. In line with this, the UK department of transport has supported various R&D projects related to the advancement of CAV [9]. Various enabling technologies have been developed to support the on-road realisation of CAV for public and private transport [10].

The early advancements in mobility planning for CAVs were majorly based on sensor-oriented predictions [11]. For example, a vision centric steering control and path planning was suggested for autonomous vehicles focusing on polynomial interpolation problem-based modelling of the on-road traffic environment [12]. A point-by-point vehicle movement plan and iterative steering supervision strategy were developed relying on on-road real-time vision or images. Towards enhancing the vision-based path planning for CAVs, manoeuvres-based path planning was suggested focusing two-level of information gathering from the on-road traffic environment [13]. In the first level, information gathering focused on the feasibility of manoeuvres considering the traffic scenario whereas in the second level optimisation of manoeuvres has been performed in terms of various performance metrics, including transportation time, rules of traffic, fuel consumption and ease of travel. For incorporating some complex steering control situations, such as sliding effects of tires, a new autonomous vehicle control system was suggested considering slip angles of the vehicle in kinematic modelling of the path following traffic environment [14]. The aforementioned advancements heavily relied on sensor data-centric mobility planning in on-road traffic environments without applying any learning from historical scenarios or data.

The recent advancement in advanced machine learning has enabled deep learning techniques to be applied in diverse domains [15,16] including mobility planning for CAVs. For example, deep neural network-based calibration automation has been suggested for improving the accuracy of sensor centric distance calculation [17]. Specifically, stereo matching and lidar projection-based path planning for autonomous vehicles were enhanced by modelling calibration network and loss function. However, the accuracy of the calibration automation approach is highly reliant on the accuracy of sensor data of autonomous vehicles without considering the overall traffic environment. Towards addressing the sensor-based path planning, a reinforcement learning-based car following model has been suggested focusing on video frame analysis [18]. In particular, you look, once the strategy has been developed, for identifying leader vehicles and obstacles in the car-following traffic environment. Q-learning and deep q-learning have been used for autonomous vehicle path planning considering red green blue depth frames in the on-road traffic video analysis. Similarly, for improving the autonomous vehicles’ consumer experience, the deep learning-based caching mechanism has been suggested focusing on edge-based media execution [19]. However, autonomous vehicle decision making was not considered in the media-oriented study. In these aforementioned machine learning and deep learning-centric CAV investigations, deep learning-enabled traffic knowledge sharing is lacking among autonomous vehicles in the same traffic environment.

Towards this end, this paper presents a deep learning-enabled framework TAKEN- Traffic Knowledge sharing based Navigation for connected and autonomous vehicles. Different autonomous aspects have been considered such as state estimation, visual perception and path or motion planning for effectively reducing the dependency of autonomous vehicles on the existing navigation systems. The major contributions of the paper can be summarised as follows:

The rest of the paper is organised as follows. In Section 2, related literature on CAV is critically reviewed. Section 3 presents the modelling detail of the proposed framework TAKEN. Section 4 discusses experimental results and analysis followed by a conclusion and future work in Section 5.

## 2. Related Work

The advancements in mobility planning for CAVs can be divided into two major sub-themes including sensor-enabled predictions and advanced machine learning enabled predictions [20,21]. Towards sensors enabled predictions, a vision centric steering control and path planning was suggested for autonomous vehicles focusing on polynomial interpolation problem-based modelling of the on-road traffic environment [12]. A point-by-point vehicle movement plan and iterative steering supervision strategy were developed relying on on-road real-time vision or images. However, vision centric steering control is limited in terms of overall traffic knowledge-centric movement planning considers not only neighbouring vehicles rather whole traffic environments in a larger area. Similarly, manoeuvres-based path planning was suggested focusing on two levels of information gathering from the on-road traffic environment [13]. In the first level information gathering focused on the feasibility of manoeuvres considering the traffic scenario whereas in the second level optimisation of manoeuvres has been performed in terms of various performance metrics including transportation time, rules of traffic, fuel consumption, and ease of travel. However, manoeuvres-based path planning is lacking real-time efficiency criteria considering level-wise execution as some sensors’ input needs to process immediately rather than waiting for other sensors to calibrate. A new autonomous vehicle control system was suggested considering slip angles of vehicles in kinematic modelling of the path following traffic environment [14]. However, the slip angle-based framework is limited in terms of applicability in sparse traffic environments due to the dependence on neighbouring vehicles.

Towards advanced machine learning-enabled predictions, deep neural network-based calibration automation has been suggested for improving the accuracy of sensor centric distance calculation [17]. Specifically, stereo matching and lidar projection-based path planning for autonomous vehicles were enhanced by modelling calibration network and loss function. However, the accuracy of the calibration automation approach is highly reliant on the accuracy of sensor data of autonomous vehicles without considering the overall traffic environment. A reinforcement learning-based car following model has been suggested focusing on video frame analysis [18]. Here, you look, once the strategy has been developed, for identifying leader vehicles and obstacles in the car-following traffic environment. Q-learning and deep q-learning have been used for autonomous vehicle path planning considering red, green and blue depth frames in the on-road traffic video analysis. Similarly, a deep learning-based caching mechanism has been suggested focusing on edge-based media execution [19]. The multi-sensor data-fusion algorithm has been developed using unscented Kalman filter to compute improvised Unmanned Surface Vehicle (USV) operation in a practical environment [22]. The robust fuzzy sliding mode rule has been applied to guide the Autonomous Underwater Vehicles (AUV) [23]. Here, both sensors enable path planning and machine learning-enabled path planning [24,25] learning from historical scenario data and traffic knowledge sharing is lacking among autonomous vehicles in the same traffic environment which is the core target of this research detailed in the following sections.

## 3. TAKEN-Traffic Knowledge-Based Navigation for CAVs

Overview of the proposed TAKEN framework is presented in Figure 1 highlighting major components. The workflow of the framework begins with the self-driving agent discerning its traffic environment by capturing data from various in- and out-bound sensors. The raw sensor data are processed by the analysis module to generate accurate control inferences. The Knowledge Sharing module improves resource utilisation of the autonomous vehicle. The modelling details of the TAKEN system are presented in the following subsections.

### 3.1. Traffic Knowledge Generation-Analysis Module

Localization and environment perception are one of the most critical tasks while piloting a vehicle [26,27]. The analysis module, as mentioned earlier, is responsible for cleaning the raw information acquired by sensors and generating control commands. The control commands are generated based on the events encountered by an agent [28,29,30]. These control commands are a consequence of state estimation, visual perception, low-level planning and behavioural modelling.

#### 3.1.1. Waypoints and Velocity Generator

The lateral and the longitudinal dynamics of the vehicle defines the steering angle and the throttle position respectively. Constrained with different forces/attributes such as the slip, radial forces, friction, centrifugal force, the aerodynamics and the banking of the road, the main goal of the lateral controller is to follow a predefined path such that the resultant (cross-track) error is minimised and that of the longitudinal controller is to maintain the reference velocity. The Stanley controller [7], a type of geometric controller, is used to model the lateral dynamics of the vehicle.
(1)$$\delta \left(t\right)=\Psi \left(t\right)+\frac{\left(k*e(t)\right)}{\upsilon (t)},\delta \left(t\right)\in \left[\delta min,\delta max\right]$$

Equation (Equation 1) [24] gives the corrected steering angle for the current time step. The controller, given by equation 1 combines the following requirements: elimination of heading error relative to the path, Ψ(t), elimination of the cross-track error e(t) and clipping of the resultant so that the steering angle lies between [δmin,δmax], where δmin and δmax are extremities (usually symmetric about zero, selected by the designer), with velocity υ(t) and some constant *k*. On the other hand, the PID (Proportional–Integral–Derivative) controller [31] is used to define the longitudinal dynamics of the vehicle.
(2)$$u=k_{p}\left(v_{d}-v\right)+k_{i}\int_{0}^{t}\left(v_{d}-v\right)dt+k_{d}\frac{d(v_{d}-v)}{dt}$$

Equation (Equation 2) represents the PID controller and is mathematically formulated by adding three terms dependent on the error function. These terms include a proportional term which is directly proportional to the error function, an integral term which is proportional to the integral of the error function and a derivative term which is proportional to the derivative of the error function. Here, the error function, $\left(v_{d}-v\right)$, is the difference between the desired speed (or desired velocity) and actual speed (or actual velocity). The pure gain term $k_{p}$, scales the vehicle’s acceleration, ensuring correct acceleration direction and magnitude. The integral term $k_{i}$ signifies the removal of accumulated error and the derivative term $k_{d}$ dampens the overshoot caused by the integral term. The output of the above equation is the corrected acceleration (or corrected velocity) that the system must apply to reach the desired speed (or desired velocity).

These systems work only if there is a contiguous supply of information from sensors such as the GNSS and the wheel odometers without which it would be difficult for a self-driving agent to estimate its state in the environment. The performance of these systems depends heavily upon the hardware devices used to measure them. Typically, measuring devices are subjected to noise and the data captured by such a device are not accurate, making them unreliable. Methods such as least squares, recursive least squares and Kalman filters can be used in deriving such values as expressed in the book named, “Optimal State Estimation: Kalman, H*∞*, and Nonlinear Approaches”, by Dan Simon [32]. However, they fail to capture major malfunctioning of the device as a result of which the estimated states turn out to be misleading. The TAKEN system overcomes such issues and the problems of sensor failures by establishing an AI estimator whose job is to align itself with the dynamic controllers (lateral and longitudinal controllers) of the vehicle while they are functioning properly and estimate/predict the state when they are ineffective. Neural networks are universal approximators. The TAKEN system replaces the path information between two points (the source and the destination) with a deep neural network. The work samples appropriate coordinates between two points and feed them to the neural network as input and trains to predict the heading and the velocity of a vehicle at a point. Equations (3) and (4) represent the cost function for the heading and the velocity predictor, respectively.
(3)$$headingPredictorModel=\frac{1}{2m}\sum_{i=0}^{m}{\left(F_{1}\left(x_{i},k\right)-y_{i}\right)}^{2}$$
where:

$x_{i}$ = (a,b) represents the map coordinates or latitude and longitude information, $y_{i}$ represents the heading at that point and
$$F_{1}\left(x_{i},k\right)=\left((a+k),b),((a-k),b),(a,(b+k)),(a,(b-k)\right)$$
such that $F_{1}\left(x\right)$ is the output of the function approximator and *k* is linearly spaced values between 0 and *p*, while sampling coordinates perpendicular to the direction of forwarding velocity of the car and linearly spaced values between 0 and *q* while sampling along the direction of forwarding velocity of the car. Here, *p* and *q* are the vehicle offsets such that, 0≤p≤q≤vehicleLength∗2, and *m* is the number of training instances.
(4)$$velocityPredictorModel=\frac{1}{2m}\sum_{j=0}^{m}{\left(F_{2}\left(x_{j}\right)-y_{j}\right)}^{2}$$
where:

$x_{j}$ is the current forward velocity at a given point (a,b),

$y_{j}$ represents the desired velocity at that point and

$F_{2}$(x) is the output of the function approximator and m is the number of training instances.

Computing the velocity is trivial because it can be obtained easily by the speedometer, however, calculating pseudo velocity can act as a bypass during the cases of latency and glitches while computing the throttle positions. To consider safety, we force the model to overfit. The advantage of this is that one could use the same model architecture to trace different paths and, thus, the amount of storage required to store any path information remains the same. As a result, even without using GNSS systems, the autonomous vehicle would be able to identify itself in the environment by just keeping the track of its heading and velocity. The system is implemented using a feedback loop such that at every timestep the current position and the heading of the vehicle are estimated from the previous timestep’s output. In other words, the forward velocity of the vehicle inferred from the velocity predictor model or the speedometer is used to estimate the next position and the heading (using the heading predictor model) of the autonomous vehicle which in turn guides its manoeuvring. This process is represented in Figure 2.

#### 3.1.2. Visual Perception

For any agents or objects on the road, objects and event identifications are required. Cameras are very important for an SDC because the data generated by such an instrument are very close to what humans can perceive. Therefore, different types of computations can be performed on such data to obtain valuable knowledge. Cameras are generally characterised by metrics such as field of view, resolution, dynamic range, focal length (*f*), depth of field and frame rate. Cameras are used in tasks such as depth estimation, segmentation, object detection and localisation, lane estimation, etc.

In the TAKEN system, depth information is obtained using stereo cameras with the assumption that the setup has two cameras whose focal lengths are equal and optical axes are parallel to each other. Figure 3A shows how the stereo camera is set up in an ideal case.

Figure 3B is the bird’s eye view of the stereo system. ‘Z’ is the depth and our goal is to calculate ‘Z’ along with the sector formed by the object of interest. “*X*-axis” represents the virtual screen in the 3D space. ‘O’ is the object in the real world. ‘O_l’ and ‘O_r’ are the images formed on the virtual screens of left and right RGB cameras respectively. Triangles formed by 127 and 139 are similar. Therefore,
(5)$$\frac{Z}{f}=\frac{X}{X_{L}}$$

Similarly, triangles formed by 458 and 469 are similar and
(6)$$\frac{Z}{f}=\frac{X-b}{X_{R}}$$

Figure 4 depicts the aerial view of different scenarios encountered by the self-driving agent. Figure 4A represents the case wherein overtaking is possible by the self-driving agent. Figure 4B, on the other hand, highlights a situation in which the self-driving agent has decided to overtake but encounters another vehicle approaching it and renders overtaking risky. Figure 4C shows a scenario in which a pedestrian obstructs the self-driving agent’s drive path. The green-coloured line represents the path that the car must take to reach the destination. The blue and red curves are the consequence of low-level decision making, red meaning invalid/risky path and blue depicting a safe path. The red-coloured car is the self-driving agent and others are non-players (i.e., other actors in the scene). The equivalent scenarios from the driving agent’s perspective as in the Carla simulator are shown in Figure 5.

We can calculate disparity as follows:(7)$$\mathrm{Disparity},\;d=X_{L}-X_{R}$$
where, $X_{L}=P_{l}-P_{0}$; $X_{R}=P_{r}-P_{0}$; and P = (U,V) represents a pixel in the pixel coordinate system (see Figure 3).

Therefore, we can compute depth as: (8)$$Z*X_{L}=f*X\;\&\;Z*X_{R}=\left(f*X\right)-\left(f*b\right)$$
(9)$$Z=\frac{f*b}{X_{L}-X_{R}}=\frac{f*b}{d}$$

Equation (Equation 9) gives us the depth of an object identified on the virtual screens. This is used to estimate the sector of impact with respect to an object detected in the image and is computed using Equation (Equation 10). The sector of impact is calculated as a function of camera intrinsic metrics and the angle formed by the two extreme points of the detected object. Thresholding this information can help us estimate the state of static and dynamic objects in the vehicle’s vicinity.
(10)$$pedal\_position=F(cos^{-1}\left(\frac{\vec{P}\vec{Q}}{\left|P|.|Q\right|}\right),camera\;intrinsics)$$
where P and Q are vectors of the form [x,y] such that (x,y) is the point in the image.

This is used by the behavioural model in the process of decision making. In order to detect objects in the scene, the work trains the YOLOv3 [10,12] model on the custom dataset containing labelled synthetic images of both static and non-static objects extracted from the Carla simulator.

#### 3.1.3. Low-Level Decision Making

The task of motion planning involves finding the best path from the source to the destination (high-level planning) and taking care of events such as lane changing, obeying traffic rules, identification and tracking of static and dynamic obstacles, collision detection, etc. (low-level planning). The work focuses on low-level planning by designing a prioritised behavioural model which would enable safe overtaking of vehicles, object collision avoidance, roadblocks and abidance by the traffic rules. Behaviour planning involves making high-level decisions to ensure safe driving. The behavioural models are triggered purely based on the visual perception stack’s inferences. The TAKEN system defines them as a state machine to handle pedestrians in the scene (Figure 6A), vehicles approaching the traffic light (Figure 6B) and perform overtaking or handling the situation of roadblocks (Figure 6C). Integrating these concepts helps us in designing a level three autonomous vehicle and enables manoeuvring in medium traffic. They are represented in Figure 4 and Figure 5. The case of overtaking a vehicle—Figure 6C—is highlighted in Figure 4A and Figure 4C. The self-driving vehicle uses cameras to estimate the depth of various objects in the scene and estimates their time of collision with that object. The red triangles in the aforementioned figures show the collision event boundaries for a given velocity. On one hand, the process of low-level decision making enables the agent to take alternate immediate routes in the case of obstructions but on the other hand, the Stanley controller [31] provides the corrected steering angle, enforcing the vehicle the adhere to its goal track and these put together enables safe overtaking. Figure 4C, explains one of the extreme cases of overtaking in which a self-driving agent initially decides to overtake because of the estimated feasibility but the vehicle which is being overtaken and the approaching vehicle apparently increases their velocities. In such a case the self-driving agent comes to a halt until there is enough safe space available for its manoeuvring. The same concept is extended when the obstructions are pedestrians and it is represented in Figure 4B.

### 3.2. Knowledge Sharing

Figure 7 shows the knowledge sharing module of the TAKEN system. This module shares knowledge with the newly introduced vehicles in a network to facilitate appropriate decision making for those vehicles even without processing information from their environments. This module mainly runs in the cloud with two major tasks—one, to store the knowledge acquired by different self-driving agents and two, to extract knowledge from different scenarios encountered by self-driving cars (SDCs). The SDCs provide continuous data to perceive the environment through various calibrated cameras and sensors and consistently upload the processed information to the cloud. To achieve knowledge sharing, the raw information acquired by sensors are classified into various possible events that an SDC would face while driving.

As a result of the training process, the function approximator is associated with parameters tuned to recognise different events. Every SDC in the network bears a database which would store the tuned parameters, the low dimensional representation of scenarios along with the decision that it took and the corresponding outcome. Although the cloud stores a combined copy of such databases across the network, autonomous vehicles in the network can access the processed information from other vehicles. This avoids requesting and packet jamming at the cloud node. To account for reliability, the architecture enables the client (SDC)—cloud interaction for the decision-making process, only if the network latency is beyond a certain threshold. Therefore, the task of decision making reduces to interpreting the scenario as a low dimensional vector using parameters stored in its database and deducing decisions by communicating in the network. The cloud periodically updates the status of the database across the network and fine-tunes the model so as to keep it informed. The advantage of this architecture is that the computational power and the time required for processing data and the amount of data to be stored in an SDC are reduced. This is depicted in Figure 8. The unused part of computational resources can be used as subordinates to the cloud which would help in better performance.

### 3.3. CAV Controllers

The controllers receive control commands from both the knowledge sharing module and the analysis module of the autonomous vehicle. However, if the response is received by the knowledge sharing module before the analysis module finishes its execution and the response of the knowledge sharing module is associated with high confidence, an event destruction trigger is passed to the analysis module to abort its current execution and prepare resources for the next set of events. Optionally, the controllers can also override/reject the output of the knowledge sharing module and continue to wait for the analysis module to complete its execution. This set of control commands is responsible for the movement of the vehicle. As a result of the actions taken by the autonomous vehicle in the environment, the responses and events are sensed/recorded by the various onboard sensors and fed back to the analysis module.

## 4. Experiments

The TAKEN system uses the Carla Simulator to simulate the autonomous vehicle and its environment. It is an open-source tool, transparent and closely related to the real world. It supports the simulation of different sensors such as LIDAR, RADAR, GNSS system, odometer, depth cameras, segmentation cameras and RGB cameras. People can build environments using the Unreal engine and associate rules and regulations. Carla simulator provides free access to its various digital assets such as vehicles, pedestrians, buildings and other entities present in the scene. The simulator is characterised by scalable client-server architecture. It also enables the programmers to attach more than one player to the same server. In addition, it facilitates traffic management, recording and tracing, ROS bridging and Autoware implementation, scenario runners and public contributions.

The Carla simulator provides us with a variety of towns/maps in which we could test our autonomous models. The working environment registers a set of actions taken by the player and associates with it the corresponding responses. The player on the other hand perceives its environment through various inbuilt simulated sensors and uses it as the source/input for the next action that ought to be taken, this is represented in the Figure 9. The environment is also associated with non-players such as pedestrians and other vehicles. The TAKEN system seeds such instances and simulates their movement/actions so as to align them with the behavioural models.

The TAKEN system implements a simulated car which is capable of driving on its own in the Carla simulator’s environment. The system uses several functionalities and add-ons provided by Carla. The proposed model falls under level three autonomy. Different situations and scenarios such as the sudden appearance of a pedestrian or a vehicle in front of the car, roadblocks, overtaking, and roundabouts which we generally come across while driving, are addressed in this work using various concepts namely, visual perception, state estimation, planning and controllers. The simulated player can manoeuvre correctly by adhering to traffic rules and regulations in medium traffic conditions. In addition to these, the paper elaborates on strategies through which we could account for sensor failures, navigation systems specific, and the task of knowledge sharing using the centralised/decentralised communication architecture. Table 1 delineates the uniqueness and the general qualitative comparison of the TAKEN system with other existing works.

## 5. Results

Figure 10A,B represent the implication of using the Stanley controller on the simulated car model. This module makes use of the navigation system to identify its current position in the environment and thus estimates the steering angle. Figure 10A represents a set of waypoints that a model must take to reach the destination and Figure 10B represents the path taken by the simulated car guided by the Stanley controller, both in terms of the map coordinate system. The effect of using the PID controller is shown in Figure 10E. The red-coloured curve represents the reference speed and the blue-coloured curve represents the actual speed measured with respect to simulator time in seconds vs speed in kmph. In case of sensor failures or erroneous sensors, the control shifts to the AI state estimator which is used for the identification of the current position in the environment. The aforementioned modules are run in parallel so as to keep them synchronised. Figure 10C,D are the outputs of the AI state estimator obtained during the execution. Figure 10C represents the velocity that a vehicle must use as a reference to define the throttle position. On the other hand, Figure 10D shows how the vehicle abides by the designated path. Here the blue points represent the predicted value and red points represent the actual/true value. This is the normalised representation of the Carla environment whose axes are longitude, latitude and velocity/heading respectively.

The knowledge sharing module in the TAKEN system uses images as raw information for detecting potholes on the road (Figure 10F) and cautioning autonomous vehicles about road conditions. Figure 10G shows that the autonomous vehicle maintained the correct lane during its course of action. For every time step, if the vehicle maintained the correct lane, the agent is rewarded with value one, otherwise, it is credited with zero rewards. In the execution environment, during the events such as overtaking and intersection crossing, the rewards are considered to be zero. Pretrained Convolution Neural Network (CNN) models such as VGG19, Xception, InceptionV3, MobileNet, ResNet152 and ResNet152V2 were used to identify events. Figure 10H represents the accuracy of each model used to design the knowledge sharing module. It is observed that all of these models performed well on the dataset. The main intention of using the model is only to attain a low dimensional representation of an event by which we could calculate similarities between different events so as to make decisions. However, information accumulated over time enables us to perform better and is considered a future work.

Figure 10I depicts the model’s loss during training for two thousand five hundred iterations in detecting and recognising objects on our custom dataset as a part of the visual perception stack. The resultant model achieved an accuracy of 93% on this dataset. Figure 11 represents the estimation of depth and sectors for every object detected in the frame. For any detected object whose depth is less than 15 meters, the autonomous vehicle performs the sector analysis so as to identify where the object of interest lies in the scene and depending on this, corresponding actions are taken by the autonomous vehicle. It is observed that the vehicle comes to a halt when an object is detected in the vehicle’s threshold region. Figure 12 is the sample output of the knowledge sharing module which illustrates the idea of knowledge sharing between the cloud and players for the identification of good and bad conditioned roads. Table 1 delineates the uniqueness and the general qualitative comparison of the TAKEN system with other existing works. Thus, in this paper we have elaborated on strategies through which we could account for sensor failures, navigation systems specific, and the task of knowledge sharing using the centralised/decentralised communication architecture.

## 6. Conclusions

The TAKEN system has implemented various components of an SDC namely, visual perception, state estimator, motion planning and behavioural modelling, using the Carla simulator. The TAKEN system has successfully devised an alternative solution to handle issues faced by self-driving agents due to sensor failures. In addition to this, the work has also introduced how one could infer knowledge from other players in the environment so as to make wiser and quicker decisions. The main motivation behind introducing this concept was from papers such as “Crazyswarm: A large nano-quadcopter swarm” [33] in which the quadcopters collectively estimate state and performs trajectory planning. The drawback of this work is that the knowledge-sharing module has been trained only on a subset of a class—pothole images. Because the training is performed with images and not videos, historical information about series of actions and situations has not been accounted for. We look forward to improving the current work’s perception stack by considering these issues. We plan on achieving this by extracting the information from the latent space of the time-series-action-pair data (that is, a pair of the situation and the corresponding action over time) and estimating the expected action of the player. Deep Reinforcement learning is an unsung branch of artificial intelligence with the potential to help us implement a solution for this problem. This concept will allow the model to learn on its own by exploration and exploitation. We will also be working on scaling up the communication architecture discussed in the TAKEN system and methodology section by providing additional functionalities so that it can aid vehicles in different complex scenarios. Building accurate models with minimal resource consumption is an open challenge in this work. The result of this work is recorded as a video and can be viewed by opening the following custom link. https://drive.google.com/file/d/1IXyGhBM2OLqZS4HTRtfFpyoeYW-f11aI/view?usp=sharing (accessed on 30 October 2022).

## Figures and Tables

**Figure 1 sensors-23-00653-f001:**
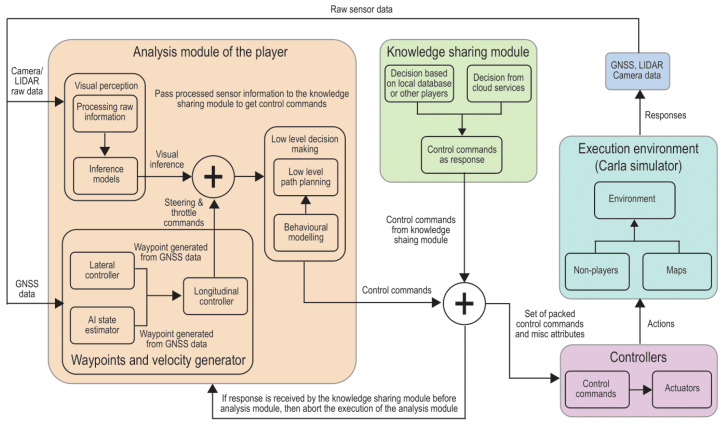
Architecture of the proposed TAKEN system.

**Figure 2 sensors-23-00653-f002:**
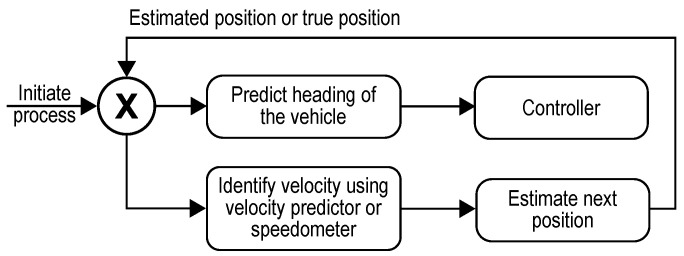
Handling cases of sensor failure.

**Figure 3 sensors-23-00653-f003:**
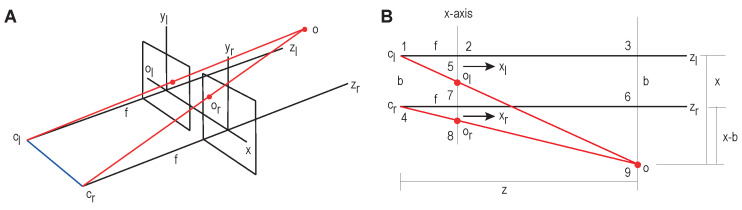
Stereo system setup showing the stereo camera setup in (**A**) and a bird’s eye view of stereo system in (**B**).

**Figure 4 sensors-23-00653-f004:**
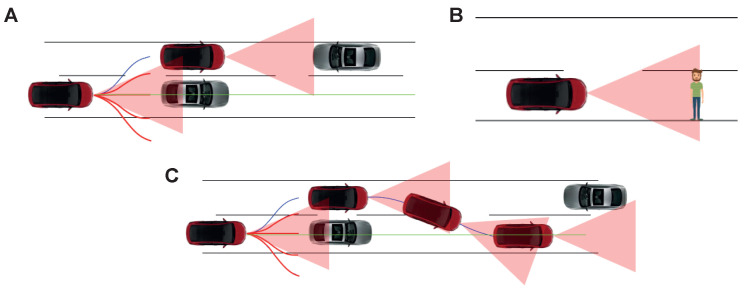
Three example scenarios (**A**–**C**) with their aerial views.

**Figure 5 sensors-23-00653-f005:**
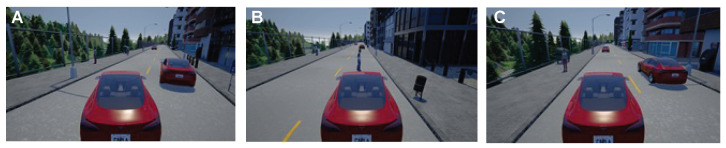
Three example scenarios (**A**–**C**) self-driving agent’s perspective.

**Figure 6 sensors-23-00653-f006:**
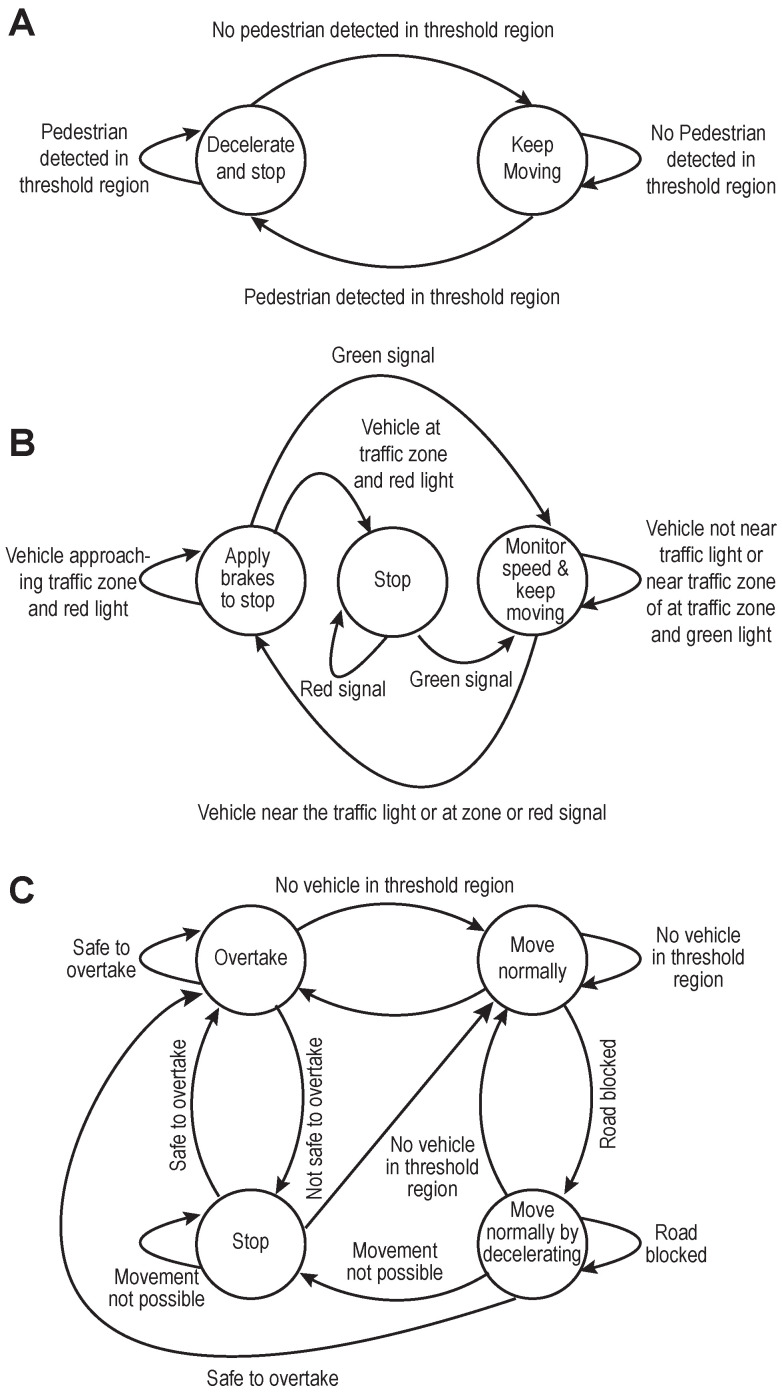
Transition diagrams for handling different traffic situations. The transition diagram for: handling pedestrians is shown in (**A**), vehicles approaching a traffic light is shown in (**B**) (yellow and red lights are treated as the same colour), and normal road condition and overtaking is shown in (**C**).

**Figure 7 sensors-23-00653-f007:**
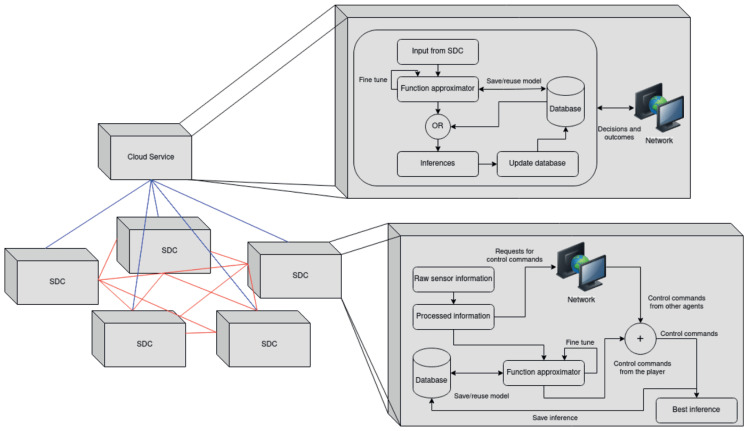
Centralised and decentralised structure of the knowledge sharing module (SDC: Self-Driving Car).

**Figure 8 sensors-23-00653-f008:**
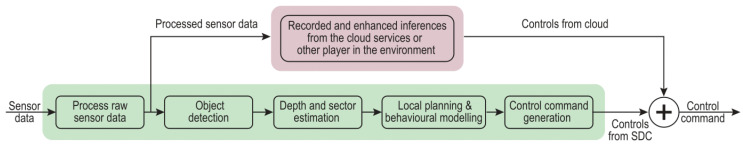
Bypassing the processing stack by making a request to the clouds or other agents in the network for the control commands corresponding to the events encountered by the autonomous vehicle.

**Figure 9 sensors-23-00653-f009:**
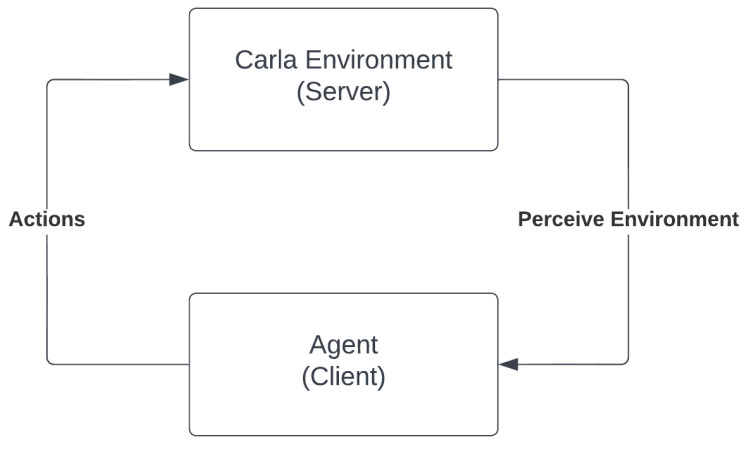
Environment-agent interaction (here, the environment is the Carla Simulator Server and the agent is a client side python script modelling SDC).

**Figure 10 sensors-23-00653-f010:**
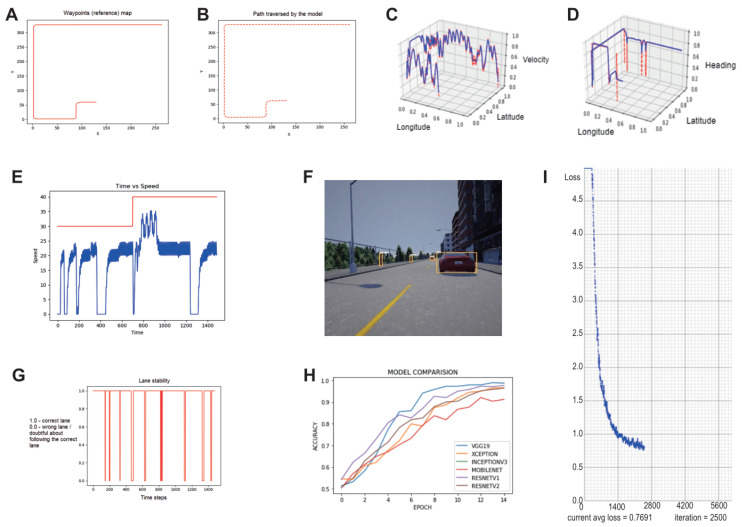
Overview of the testing and the corresponding results. The reference path is shown in (**A**) and the traversed path is shown in (**B**). The velocity and heading predictors have been shown in (**C**) and (**D**), respectively. The effect of using a PID controller in the designed environment is shown in (**E**) (max speed limited to 30 kmph). The output of the object detection is shown in (**F**) and the lane stability graph in (**G**). The accuracies of the different machine learning models are shown in (**H**) with the average loss of the models for custom object detection in (**I**).

**Figure 11 sensors-23-00653-f011:**
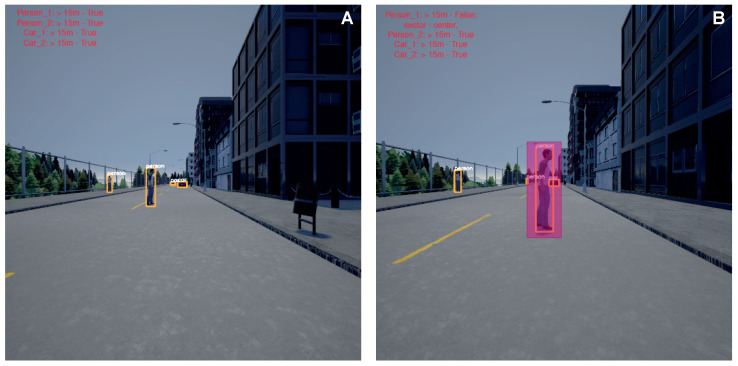
Estimation results by the TAKEN model with object and depth estimation in (**A**) and depth and sector estimation in (**B**).

**Figure 12 sensors-23-00653-f012:**
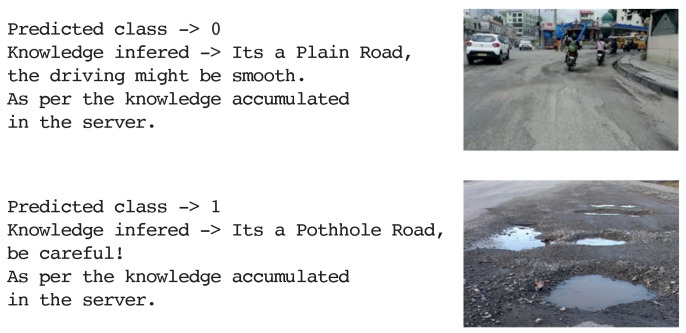
Road condition estimation.

**Table 1 sensors-23-00653-t001:** Comparison of the TAKEN system with other existing works.

Key Concepts	Existing Works	Uniqueness of the TAKEN System
AI State Estimator	It was observed that methods such as RLS and Kalman filters were used to handle erroneous sensors but not their failures. The main advantage of these methods is that they compute the true value of any quantitative entity on the fly.	Usage of neural networks to define paths between any two points and using heading and velocity predictor to estimate its current state in the cases of sensor failure or erroneous sensors. This method enables a system to move between two points without a navigation system.
Visual Perception	Most of the existing works used pretrained models trained on Coco/ImageNet dataset to detect objects in the scene.	The TAKEN system uses a custom dataset containing annotations for pedestrians, vehicles, traffic lights and signals and other static objects, defined in the synthetic environment of the Carla simulator.
Knowledge Sharing Module	This ideology is absent in most of the existing work.	The work proposes a centralised–decentralised architecture for communicating among other agents and cloud services in the environment to enhance the process of decision making and reduce computation requirements.

## Data Availability

Not applicable.

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
