# Peer review of "TAKEN: A Traffic Knowledge-Based Navigation System for Connected and Autonomous Vehicles"

_sensors, 2023, doi:10.3390/s23020653_

Round 1

Reviewer 1 Report

The authors propose a model for mining the sensor-oriented traffic data to generate a precise navigation path for the vehicle. A knowledge-sharing method is developed for collecting and generating new traffic knowledge from on-road vehicles. The paper is overall well written. 

1. Add chapter Experiments in which you describe the experiments presented in the chapter Results. You can use some of the text that is currently in Chapter 4. You could create a graphical scheme of the experiment plan. That way, the paper will be easier to follow. 

2. Create a separate chapter for the Results. Please provide more information about the results. The current information is too short for the reader to conclude the proposed system. Compare it with other systems, and provide some background on why the proposed system is relevant for practice and theory. 

3. The current conclusion is too short. In the last section, please focus on “Discussion, Implication, and Conclusion” to include

(1).     Summary of the research - what was the goal, and how was it attained (2).     Discussion of why the authors found these results and how they comply (or do not) with the Literature Review. (3).     Managerial Implications (4).     Limitations of the paper (5).     Future Studies and Recommendations  

Author Response

The authors propose a model for mining the sensor-oriented traffic data to generate a precise navigation path for the vehicle. A knowledge-sharing method is developed for collecting and generating new traffic knowledge from on-road vehicles. The paper is overall well written.

Response to the Reviewer: Thank you for the comment.

  1. Add chapter Experiments in which you describe the experiments presented in the chapter Results. You can use some of the text that is currently in Chapter 4. You could create a graphical scheme of the experiment plan. That way, the paper will be easier to follow.

Response to the Reviewer: Thank you for the valuable comment. We have now added chapters Experiments and Results and set the content of the paper appropriately. The graphical scheme has been already shown in Figure 1 which gives a quick overview of the proposed work in terms of various modules used.

  1. Create a separate chapter for the Results. Please provide more information about the results. The current information is too short for the reader to conclude the proposed system. Compare it with other systems, and provide some background on why the proposed system is relevant for practice and theory.

Response to the Reviewer: Thank you for the valuable comment. We have now added a separate chapter for the Results section. The Results section has been reframed inorder for the reader to conclude the proposed system. Table 1 gives the comparison of the TAKEN system with other existing works.

.

  1. The current conclusion is too short. In the last section, please focus on “Discussion, Implication, and Conclusion” to include:

(1).     Summary of the research - what was the goal, and how was it attained (2).     Discussion of why the authors found these results and how they comply (or do not) with the Literature Review. (3).     Managerial Implications (4).     Limitations of the paper (5).     Future Studies and Recommendations 

Response to the Reviewer: Thank you for the valuable comment. We have now modified the conclusion section appropriately.

Reviewer 2 Report

-          Today, autonomous studies have started to be used in every sector. There are widespread studies on this subject in the highway, seaway, railway and airway sectors. In the introduction part of the study, it will be useful to present sample studies made in these sectors. If possible, this can be presented to the reader in tabular form. As a seafarer, I know a lot of work done in maritime transport. In this table, it would be appropriate to include the following studies while giving place to the studies carried out in the sectors:

-          Liu, WWLiu, YC and Bucknall, R (2022). Filtering based multi-sensor data fusion algorithm for a reliable unmanned surface vehicle navigation. JOURNAL OF MARINE ENGINEERING AND TECHNOLOGY. DOI: 10.1080/20464177.2022.2031558

-          Lakhekar, GV and Waghmare, LM. “Robust self-organising fuzzy sliding mode-based path-following control for autonomous underwater vehicles”, JOURNAL OF MARINE ENGINEERING AND TECHNOLOGY. Indexed: 2022-09-21, DOI: 10.1080/20464177.2022.2120448

Please add explanatory information about all abbreviations in the equations in the study.

Comparing the results of the study with studies from the literature was insufficient. This part needs to be strengthened.

Author Response

Today, autonomous studies have started to be used in every sector. There are widespread studies on this subject in the highway, seaway, railway and airway sectors. In the introduction part of the study, it will be useful to present sample studies made in these sectors. If possible, this can be presented to the reader in tabular form. As a seafarer, I know a lot of work done in maritime transport. In this table, it would be appropriate to include the following studies while giving place to the studies carried out in the sectors:

-          Liu, WW; Liu, YC and Bucknall, R (2022). Filtering based multi-sensor data fusion algorithm for a reliable unmanned surface vehicle navigation. JOURNAL OF MARINE ENGINEERING AND TECHNOLOGY. DOI: 10.1080/20464177.2022.2031558

-          Lakhekar, GV and Waghmare, LM. “Robust self-organising fuzzy sliding mode-based path-following control for autonomous underwater vehicles”, JOURNAL OF MARINE ENGINEERING AND TECHNOLOGY. Indexed: 2022-09-21, DOI: 10.1080/20464177.2022.2120448.

Response to the Reviewer: Thank you for the valuable comment. We have added the literature on Unmanned Surface Vehicle and Autonomous Underwater Vehicles. Kindly refer reference 27 and 28. Due to the length of the paper, we have not included the survey again in a tabular form as the detailed survey is provided in section 2.

  1. Please add explanatory information about all abbreviations in the equations in the study.

Response to the Reviewer: Thank you for the valuable comment. The abbreviations are included wherever necessary. A general abbreviation section is included at the end of the paper.

  1. Comparing the results of the study with studies from the literature was insufficient. This part needs to be strengthened..

Response to the Reviewer: Thank you for the valuable comment. Comparison of the TAKEN system with other existing works is given in Table 1. Knowledge Sharing Module is absent in most of the existing work and this is the unique contribution of this work.

Round 2

Reviewer 2 Report

accept